# OpenReview forum: "Uncovering Safety Risks of Large Language Models through Concept Activation Vector"
_NeurIPS.cc/2024/Conference — NeurIPS 2024 poster_

### Official Review · Reviewer_xmcv · 2024-07-11

**Soundness:** 3
**Presentation:** 3
**Contribution:** 3
**Rating:** 4
**Confidence:** 5

**Summary:**

The paper introduces a Safety Concept Activation Vector framework designed to identify and exploit vulnerabilities in LLMs by accurately interpreting their safety mechanisms. The authors develop an SCAV-guided attack method that enhances the success rate of generating harmful prompts and embedding-level attacks with optimized perturbations, requiring less training data.

**Strengths:**

1. The paper is well-written and easy to follow.
2. The evaluation is comprehensive and the results look promising.
3. The discussion on section 4.3 is insightful.

**Weaknesses:**

The idea of inspecting the hidden representation of the model about the safety concept is not novel, which is already proposed in RePE. The difference is that the authors propose to use the SCAV to find the subspaces for benign and harmful prompts, while RePE uses the PCA. The results in Figure 3 look confusing. When only one pair of benign-harmful instruction is presented in the training data, the success rate of the SCAV is already over 80%. It is not clear why it can learn the subspaces so well with only one pair of data. With only one pair of data, the subspace boundary should be very uncertain. The authors should provide more explanation on this.

The second concern is that directly modifying the intermediate representation of the model may lead to the model's performance degradation. Modifications to any internal layer will influence all the following layers' representations[1,2]. Without drift control, it is hard to guarantee that the model's performance will not be affected. The authors should provide more discussion on this. Also, the langauge flaw determined by GPT-4 is not a good metric to reflect that. I would suggest using more principled metrics such as weighted average of bi- and tri-gram entropies.

[1] Locating and Editing Factual Associations in GPT

[2] MASS-EDITING MEMORY IN A TRANSFORMER

Other issues:

Although the paper claims that they have done comfort for annotators, I think the IRB approval or equivalent is still needed since it involves human subjects and may do potential harm to the annotators.

**Questions:**

The transferability to GPT-4 is questionable, since the embedding editing only works for the local model while GPT-4 has a definitely different embedding. Could you give an reasonable explanation for that?

**Limitations:**

Yes

---

> ### Author Rebuttal · Authors · 2024-08-06
>
> Thanks for your insightful and detailed comments, which enable us to better clarify our contributions and ethical considerations.
>
> > **Weakness 1: Interpretability used in attacks by [1]**
> >
>
> We acknowledge that previous works like [1] have explored the use of interpretability to assist attacks. In contrast, our work distinctively focuses on developing a **more accurate and principled interpretation method**. Unlike existing approaches, our method:
>
> 1. Eliminates misleading heuristics (L128-140) and thus performs significantly better than existing techniques [1,2] (Tables 1 and 2), particularly under conditions of limited training data (Figure 3).
> 2. Is the first to support both embedding-level and prompt-level attacks, enhancing its applicability.
> 3. Removes the necessity for time-consuming hyperparameter tuning.
>
> We will update our abstract and introduction to better highlight these distinctions.
>
> > **Weakness 2: Confusion in Figure 3**
> >
>
> We have double-checked the result in Figure 3 and confirmed it is correct. One pair of data can lead to high performance because embeddings of malicious questions are well separated from those of safe questions (Rebuttal Table 1). In this case, we can learn a linear classifier **$P_m$** that accurately distinguishes embeddings of malicious questions from safe ones with one pair of data (Rebuttal Table 2) and use **$P_m$** to effectively guide the attacks.
>
> **[Rebuttal Table 1] Distances between embeddings of malicious and safe questions ($d_{ms}$) exceed those within malicious ($d_{mm}$) or safe questions ($d_{ss}$) in LLaMA-2-7B-Chat 31th layer.**
>
> | Type | Min | Max | Mean | Medium |
> | --- | --- | --- | --- | --- |
> | $d_{mm}$ | 11.3 | 93.3 | 56.2 | 57.7 |
> | $d_{ss}$ | 30.4 | 128.8 | 84.9 | 84.2 |
> | $d_{ms}$ | 83.0 | 132.6 | 113.9 | 114.2 |
>
> **[Rebuttal Table 2] Test accuracy of linear classifier $P_m$ using one pair of training data (LLaMA-2-7B-Chat)**
>
> | Runs | Layer 10 | Layer 15 | Layer 20 | Layer 25 | Layer 30 |
> | --- | --- | --- | --- | --- | --- |
> | 1 | 60.7 | 92.4 | 96.6 | 94.2 | 94.3 |
> | 2 | 79.1 | 96.4 | 96.3 | 96.8 | 95.9 |
> | 3 | 82.3 | 97.5 | 96.0 | 96.6 | 95.8 |
> | 4 | 86.7 | 95.6 | 96.6 | 96.3 | 93.2 |
> | 5 | 70.4 | 97.1 | 93.7 | 95.1 | 94.5 |
> | Variance | 107.27 | 4.13 | 1.49 | 1.23 | 1.27 |
>
> > **Weakness 2: Risks in modifying intermediate representations**
> >
>
> We carefully ensure such risks are under control by:
>
> 1. **Optimizing with drift control**. Our goal minimizes perturbation magnitude $|\epsilon|$ to avoid significantly modifying intermediate representations (Eq. (3)). We also modify only the necessary layers (Algorithm 1).
> 2. **Performance validation**. In addition to our **human evaluation** (Table 2), which confirms that LLMs modified by our method have good performance, we also implement **the weighted average of bi- and tri-gram entropies** per your suggestion. This quantity drops if the generated text is repetitive. As shown in Rebuttal Table 3, our method consistently performs the best in terms of this criterion.
>
> **[Rebuttal Table 3] Comparing our embedding-level attacks with baselines in terms of the weighted average of bi- and tri-gram entropies
>  [3, 7]**
>
> | Model | Method | Entropy on Advbench | Entropy on StrongREJECT |
> | --- | --- | --- | --- |
> | LLaMA-2-7B-Chat | JRE | 14.65 | 13.34 |
> |  | RepE | 16.49 | 15.33 |
> |  | DP | 15.51 | 15.69 |
> |  | AutoDAN | 16.46 | 15.62 |
> |  | SCAV (Ours) | 16.97 | 16.24 |
> | LLaMA-2-13B-Chat | JRE | 16.59 | 15.94 |
> |  | RepE | 16.54 | 15.81 |
> |  | DP | 14.58 | 15.25 |
> |  | AutoDAN | 13.03 | 12.41 |
> |  | SCAV (Ours) | 16.98 | 16.35 |
>
> > **Weakness 3: Ethical considerations in human evaluation**
> >
>
> We take the following steps to adhere to ethical standards:
>
> 1. **Obtaining IRB-equivalent approval.** Our human annotation experiment was carried out through a vendor company that has a formal contract with our institution. We have carefully confirmed with our collaborator (level 3 program manager) in the vendor company, and she has investigated our annotation process and confirmed it has passed their ethical approval.
> 2. **Doing comfort for the annotators.** We reviewed the labeling content to ensure that there was no extremely harmful content, warned the annotators about the potential harm, and let them know that they could quit at any time if uncomfortable. We also conducted a follow-up with the annotators, and they all indicated that our experimental content had not caused harm to them.
>
> > **Question 1: Transferability to GPT-4**
> >
>
> The transferability of attacks observed in our experiments is also reported in multiple existing works [4,5,6], enhancing the credibility of our results. For example, it has been found that attacks learned in LLaMA-2 can be effectively applied to GPT-3.5, GPT-4, and Vicuna [4,5,6]. Currently, to the best of our knowledge, no existing works can clearly explain its reason. One hypothesis is that there is a form of cross-model generalizability potentially resulting from using a similar architecture (Transformer), loss function (e.g., next-token-prediction), and datasets (human-written text from the Internet), and we consider the verification of this hypothesis to be an interesting future work.
>
> [1] Representation engineering: A top-down approach to ai transparency
>
> [2] Open the Pandora's Box of LLMs: Jailbreaking LLMs through Representation Engineering
>
> [3] A Unified Framework for Model Editing
>
> [4] Universal and transferable adversarial attacks on aligned language models
>
> [5] AutoDAN: Generating stealthy jailbreak prompts on aligned large language models
>
> [6] AdvPrompter: Fast adaptive adversarial prompting for LLMs
>
> [7] MASS-EDITING MEMORY IN A TRANSFORMER

---

> > ### Author Response · Authors · 2024-08-13
> >
> > Dear Reviewer xmcv,
> >
> > Thank you for your thoughtful feedback on our submission. We kindly remind you to review our rebuttal and let us know if it adequately addresses your concerns. If you believe our explanations and revisions have satisfactorily resolved the issues, we would greatly appreciate it if you could reconsider your evaluation of our paper.
> >
> > Thank you again for your time and for providing valuable guidance. If there are any further questions or suggestions, we are glad to discuss them with you.
> >
> > Best regards,
> >
> > Authors of paper 7016

---

> > > ### Comment · Reviewer_xmcv · 2024-08-13
> > >
> > > Dear Authors,
> > >
> > > Thank you for your rebuttal on your paper. While I appreciate your efforts to clarify the contributions and methodologies, I still have some concerns that prevent me from raising my score. Below are the reasons:
> > >
> > > **Novelty and Superiority of SCAV**: The use of interpretability in understanding the hidden representation of safety concepts is not novel, as similar approaches have been explored in previous works like RePE. Your paper claims that SCAV is superior to PCA or other clustering methods. However, there is no statistical analysis provided that demonstrates SCAV's superiority over these methods. Without such evidence, it is difficult to convincingly argue that SCAV offers a significant improvement or new insights over existing techniques, or it is a general discovery that could generalize to other fileds related with hidden representation inspection. Since the separation of harmful and harmless prompt representations is already a known phenomenon, this aspect of your work does not bring new insights to the community.
> > >
> > > **One-Pair Classification**: The claim that a single pair of training data can achieve high classification accuracy raises further questions about the novelty and contribution of your approach. If representations are already well-separated, as demonstrated by your one-pair classification, it suggests that any clustering method could achieve similar results. This undermines the argument that SCAV is necessary for superior performance and conflicts with the claim of SCAV's distinct advantage.
> > >
> > > **Transferability Without Explanation**: While it is recognized that other works have attempted to transfer white-box attacks to black-box models, these claims often lack a sound explanation for why such transfers work. For instance, while GCG reports good transfer performance, subsequent works like AutoDAN show very low ASR for transfer GCG. Your paper should compare SCAV with existing black-box methods for jailbreaking and provide a deeper explanation of why the transferability of your method to models like GPT-4 is expected or reliable. Without this, the work does not provide new insights into the understanding of transferability for the community.
> > >
> > > Given these concerns, I believe that your work does not sufficiently address these issues to warrant an increase in my evaluation score.

---

> > > > ### Author Response · Authors · 2024-08-14
> > > > **Clarify factual misunderstandings in your comment [1/2]**
> > > >
> > > > Thanks for your detailed feedback. We appreciate your efforts in clarifying your concerns. We would like to take this opportunity to clarify some points that may have been misunderstood.
> > > >
> > > > ### **Novelty and Superiority of SCAV**
> > > >
> > > > You said that "there is no statistical analysis provided that demonstrates SCAV's superiority over these methods." We have demonstrated SCAV’s superiority over existing methods by conducting multiple experiments, and some of experiments are conducted based on your suggestion. Here is a summarization of statical analysis that we provide to demonstrate our superiority:
> > > >
> > > > - **Analysis on averaged $P_m$ decrease**: In Table 9, we compute the average accuracy of perturbation direction of existing methods, according to whether RepE successfully lowers the malicious probability $P_m$ according to the linear classifier and the percentage is only 7%-10% compared with 0.01% of our method. These can also be mutually supported by the ASR of each baseline.
> > > > - **Analysis on ASRs and Language improvement**: In Tables 1-5, we have shown that our method improves ASR over RepE by 12%-30% and JRE by 16%-34%, decreasing the percentage of language flaws by 14%-42%.
> > > > - **Analysis on performance less data**: In Figure 3, We have shown that in the case of less training samples (1-10), SCAV reached nearly 80% of ASR at 1 sample, while ASR of RepE is 0, and JRE is only 50%. ASR of SCAV reached 100% at 5 samples. And it is consistently significantly better than the performance of RepE and JRE.
> > > > - **Analysis on vector extraction difference**: In Figure 2, we analyze why RepE and JRE may cause incorrect direction extraction. These data are synthesized according to the real data dimension reduction distribution in Figures 6 and 7, and the arrows are obtained by the real vector extraction algorithms.
> > > > - **Analysis on t-SNE embedding distribution**: In Figures 6 and 7, we have shown you the distribution of malicious instructions in Advbench and the generated safety instructions embedding after t-SNE dimension reduction. The results show that linear separability is different in different layers, and the middle and late layers are more significant.
> > > > - **Analysis on which layers to perturb**: In Figures 1, 4(a), 4(b), we analyze the significance of each layer for perturbation. The linear separability of each layer does not always exist or exist independently, and it needs to be determined automatically. These analyses illustrate the advantages of SCAV, especially Algorithm 1.
> > > > - **Analysis on weighted entropy per your request**: In Rebuttal Table 3, we have shown the language flaws analysis on more principled metrics (weighted average of bi- and tri-gram entropies) per your request. The results still hold our conclusion.
> > > >
> > > > If there are further analysis that you suggested, we appreciate your specific guidance and we are glad to provide the results.
> > > >
> > > > ### **One-Pair Classification**
> > > >
> > > > You said that "any clustering method could achieve similar results", could you clarify which clustering methods you refer to? We have compared with baselines RepE and JRE (Tables 1/2 and Figure 3), and they cannot achieve similar performance even when embeddings are well separated, because:
> > > >
> > > > - Well-separation only happens for certain layers and only our method identifies these layers to perform the attack.
> > > > - **Well-separation does not automatically lead to optimal attack.** Instead, we still need to careful in finding the optimal direction that minimizes the perturbation magnitude, which requires us to understand the separation hyperplane, and this is only performed by using our method. Existing methods that heuristically select a direction cannot always guarantee this.

---

> > > > ### Author Response · Authors · 2024-08-14
> > > > **Clarify factual misunderstandings in your comment [2/2]**
> > > >
> > > > ### **Transferability Without Explanation**
> > > >
> > > > You said that "Your paper should compare SCAV with existing black-box methods for jailbreaking". We have conducted such experiments with black-box methods DeepInception [1] and AutoDAN [2]. They are presented in Tables 3/4 and GCG [3] presented in Rebuttal Tables 4/5 per the request of reviewer CRgs.
> > > >
> > > > You said that "the work does not provide new insights into the understanding of transferability for the community". New insights in this aspect is not the focus of our paper. Our goal here is to follow existing methods and show the potential usefulness of method when applied to GPT-4. In terms of insights, we have provided multiple others and reviewers 6xey, CRgs, mJhn have acknowledged as insightful:
> > > >
> > > > - "I believe this paper makes a good improvement of LLM jailbreak attacks, which could be used to further encourage future development of stronger defenses" by reviewer 6xey
> > > > - "this work fixes an obvious problem of these methods with a simple technique" by reviewer CRgs
> > > > - "offering a unique perspective on understanding and exploiting their safety mechanisms" by reviewer mJhn
> > > >
> > > > All insights we have:
> > > >
> > > > - The first to observe and report linear separable phenomenon of malicious and safe embeddings, and verify it on 8 open-source LLMs popular in community
> > > > - Uncovering safety risks by SCAV of open-source models, black-box models and adversarial training models (unlearn [4] and per the request of reviewer mJhn, RR [5])
> > > > - The proposed algorithm shows the distribution of safety concept at each layer and reflects the relationship between them
> > > >
> > > > [1] "we mainly consider the training-free and **black-box** jailbreak, which is more practical." DeepInception: Hypnotize Large Language Model to Be Jailbreaker
> > > >
> > > > [2] "these interpretable prompts generalize better to unseen harmful behaviors and transfer better to **black-box** LLMs than the unreadable ones in prior work." AutoDAN: Interpretable Gradient-Based Adversarial Attacks on Large Language Models
> > > >
> > > > [3] "the adversarial prompts generated by our approach are highly transferable, including to **black-box**, publicly released, production LLMs." Universal and Transferable Adversarial Attacks on Aligned Language Models
> > > >
> > > > [4] Eraser: Jailbreaking Defense in Large Language Models via Unlearning Harmful Knowledge
> > > >
> > > > [5] Improving Alignment and Robustness with Short Circuiting

---

### Official Review · Reviewer_mJhn · 2024-07-12

**Soundness:** 3
**Presentation:** 3
**Contribution:** 3
**Rating:** 4
**Confidence:** 3

**Summary:**

This paper introduces the Safety Concept Activation Vector (SCAV) framework, which guides attacks on LLMs by interpreting their embedding-level safety mechanisms. This work estimates the likelihood that an embedding is considered malicious by the LLM, utilizing a linear classifier based on Concept Activation Vector principles. The attack method generates both attack prompts and embedding-level attacks with automatically selected perturbation hyperparameters. The method improves attack success rates and response quality while requiring less training data. The experimental results show that it's possible to transfer attack prompts across different models, including black-box ones like GPT-4. The findings suggest that existing unlearn methods may not fully erase harmful knowledge, highlighting the persistent safety issues in LLMs.

**Strengths:**

The paper's originality lies in its application of the SCAV framework to guide attacks on LLMs, offering a unique perspective on understanding and exploiting their safety mechanisms. The quality of the work is high, supported by comprehensive experiments that validate the effectiveness of the SCAV framework across multiple models and datasets.

**Weaknesses:**

One significant weakness of the paper is its reliance on the assumption of linear separability, which may not consistently hold true in the complex, high-dimensional spaces characteristic of LLM embeddings. The authors were correct to cite many papers on interpretability that also assume the linear interpretability assumption but to me this is a generally unsound assumption in the interpretability community. In section 2.2 you empirically evaluate linear interpretability with Llama and Vicuna, but while the results show high test accuracy in later layers, the dynamics of how linear separability develops across layers could be explored in more depth. The intuition in 2.3.1 is an oversimplification of high-dimensional spaces. Each layer in a deep neural network applies a non-linear transformation to the data. While later layers might exhibit some linear separability due to the network's hierarchical feature learning, without a mathematical explanation it is not clear to me why non-linear transformations and complex concepts would be accurately captured linearly.

The regularization terms in the objective function ( \( \frac{\lambda_1}{2}||w||^2 + \frac{\lambda_2}{2}b^2 \) ) help prevent overfitting, but their choice and tuning are not discussed in detail. Improper regularization can lead to either overfitting or underfitting, affecting the classifier's ability to generalize.


While the theoretical foundation for using Concept Activation Vectors is sound, the paper does not provide sufficient empirical evidence to validate this assumption across a diverse range of models and layers. This oversight raises questions about the general applicability and robustness of the SCAV framework. The lack of validation makes it difficult to ascertain whether the linear separability assumption can be reliably applied to different LLM architectures or whether it is specific to the models and datasets used in the experiments.

Furthermore, the decision to focus attacks on layers with high test accuracy appears somewhat arbitrary and lacks justification. The authors need to provide a deeper exploration of why these layers are chosen and how this choice impacts the effectiveness of the SCAV-guided attacks. A more detailed discussion on the selection criteria for hyperparameters, such as the thresholds \( P_0 \) and \( P_1 \), would significantly enhance the robustness of the methodology. The current approach leaves the impression that the layer selection and hyperparameter tuning are based on heuristic rather than principled optimization, which undermines the credibility of the proposed framework.

The experiments are predominantly focused on a narrow set of models (e.g., LLaMA-2-7B/13B-Chat) and specific datasets (e.g., Advenbench and StrongREJECT). This limited scope constrains the generalizability of the findings.
For example, the authors utilize GPT-4 for computing 209 ASR-answer, ASR-usefulness, and Language flaws in the experiments, but not for tables 1 and 2. They could also use Harmbench (Mazeika, Mantas, et al.) as another dataset. To convincingly demonstrate the efficacy and versatility of the SCAV framework, the authors should expand their evaluation to include a broader range of LLM architectures, including those with different training paradigms.

**Questions:**

1. Could you provide more empirical evidence to support the linear separability assumption across different models and layers? This would help validate the applicability of the SCAV framework.
2. How do you determine the thresholds P0 and P1 for attacking specific layers?
3. Have you considered testing the SCAV framework on models with adversarial training or other defensive mechanisms?

---

> ### Author Rebuttal · Authors · 2024-08-06
>
> > **Weakness 1 and Question 1: Linear interpretability assumption**
> >
>
> Thanks for explaining your concern in such a detailed and constructive way. Per your suggestion, we further justify the assumption by providing:
>
> 1. **Empirical evidence across models and layers**.  As shown in Rebuttal Table 1, the test accuracy of linear classifier $P_m$ for 5 more LLMs consistently exceeds 0.9 in later layers, confirming the robustness of the assumption. Moreover, our attack method, built upon this assumption, has achieved an ASR-keyword of over 94% on 7 LLMs, demonstrating its general applicability. You can also see Figure 1 in Rebuttal PDF for details by layer.
> 2. **Further explanations**. A solid theoretical explanation for linear interpretability is a difficult task not solved by works that use the assumption (refs [20-23] in paper). Our hypothesis is **two groups of embeddings tend to be linear separable if learned to have sufficient inter-group distance and intra-group similarity.** For example, in LLaMA-2-7b-Chat, the median distances between embeddings of malicious and safe questions ($d_{ms}$) exceed those within malicious ($d_{mm}$) or safe questions ($d_{ss}$) - 113.88 vs. 56.21 and 84.89, respectively. This disparity, learned to ensure malicious questions have different outputs compared with safe questions, may form clear clusters easily to be linearly separated.
>
> **[Rebuttal Table 1] Minimal test accuracy of linear classifier** $P_m$  **across LLMs and layers**
>
> |  | Layers 0~10 | Layers 11~20 | Layers 21~31 | Layers 31~39 |
> | --- | --- | --- | --- | --- |
> | LLaMA-3-8B-Instruct | 0.43 | 0.95 | 0.97 | NA |
> | Qwen-1.5-7B-Instruct | 0.56 | 0.71 | 0.98 | NA |
> | Mistral-7B-Instruct | 0.50 | 0.82 | 0.90 | NA |
> | Deepseek-v2-Lite-Chat | 0.55 | 0.94 | 0.94 | NA |
> | ChatGLM-4-9B-Chat | 0.62 | 0.78 | 0.98 | 0.99 |
>
> > **Weakness 2**: **Choices of regularization terms**
> >
>
> The current regularization follows the default setting in sklearn and consistently performs well on varying models and datasets. We also test different regularization terms. **Rebuttal Table 2 shows that using the L2 regularization is critical, while its coefficient is not sensitive.** We will include this analysis in the revised paper.
>
> **[Rebuttal Table 2] ASR-keyword (%) w.r.t different regularization terms (Advbench, LLaMA-2-7B-Chat)**
>
> | $\lambda (\lambda_1=\lambda_2)$ | L1 | L2 |
> | --- | --- | --- |
> | 0.5 | 0 | 100 |
> | 1 | 0 | 100 |
> | 2 | 0 | 98 |
> | 3 | 0 | 100 |
>
> > **Weakness 3: Layers to attack**
> >
>
> We attack only layers with an accurate $P_m$ because our attacks are guided by $P_m$ (Eq. (3)) and **inaccurate $P_m$ leads to ineffective attacks**. This is consistent with empirical results: perturbing a layer whose test accuracy is lower than 90% results in a low attack success rate (ASR-k from 0 to 6%), while perturbing other layers leads to a consistently better result (41.1$\pm$26.2%).
>
> > **Weakness 4 and Question 2: Determining $P_0$ and $P_1$**
> >
>
> Our method works reasonably well for varying $P_0$ and $P_1$, as shown in Rebuttal Table 3. **Users can easily determine $P_0$ and $P_1$** by using the default values (0.01% and 90%) that work well for all 7 LLMs we tested, or slightly lower $P_0$ if they wish to increase ASR. An optimal $P_1$ can be found by plotting $P_1$ across layers and choosing the elbow point, but the default value already works well in practice.
>
> **[Rebuttal Table 3] ASR-keyword (%) w.r.t. varying** $P_0$ **and** $P_1$ **(Advbench, LLaMA-2-7B-Chat).**
>
> | $P_0 \backslash P_1$ | 0.85 | 0.90 | 0.95 |
> | --- | --- | --- | --- |
> | 1e-3 | 98 | 96 | 98 |
> | 1e-4 | 100 | 100 | 100 |
> | 1e-5 | 100 | 100 | 100 |
>
> **Weakness 5: More datasets and diverse LLMs**
>
> Beyond the 2 datasets and 8 LLMs already tested (Table 5), we have extended our experiments to Harmbench and 3 more LLMs with different training paradigms. Rebuttal Tables 4 and 5 show the effectiveness of our method. We also hope to highlight that 3 reviewers claim that our original experiment is “comprehensive” or “thorough.”
>
> **[Rebuttal Table 4] Results on Harmbench**
>
> |  | ASR-keyword (%) | ASR-answer (%) | ASR-useful (%) | Language Flaws (%) |
> | --- | --- | --- | --- | --- |
> | LLaMA-2-7b-Chat | 99.5 | 97.5 | 90 | 20 |
> | LLaMA-2-13b-Chat | 98.75 | 95 | 87.5 | 13.75 |
>
> **[Rebuttal Table 5] Results on 3 more LLMs on Advbench**
>
> |  | ASR-keyword (%) | ASR-answer (%) | ASR-useful (%) | Language Flaws (%) |
> | --- | --- | --- | --- | --- |
> | ChatGLM4-9b | 94 | 86 | 82 | 18 |
> | Deepseek-v2-lite-chat | 100 | 96 | 86 | 6 |
> | Gemma-1.1-7B-it | 100 | 90 | 86 | 14 |
>
> > **Question 3: Applying defensive mechanisms like adversarial training**
> >
>
> Per your suggestion, we attack models with adversarial training (Rebuttal Table 6) and other defense mechanisms (Rebuttal Table 7). The unlearn method in Sec. 4.2 can also be regarded as a defense method. Results show that **all 6 tested defense methods cannot well mitigate our attacks**.
>
> **[Rebuttal Table 6] Attacking LLMs with adversarial training [1] on Advbench**
>
> |  | Methods | ASR-keyword (%) | ASR-answer (%) | ASR-useful (%) | Language Flaws (%) |
> | --- | --- | --- | --- | --- | --- |
> | LLaMA-3-8B-Instruct-RR | W/o Attack | 0 | 0 | 0 | 96 |
> |  | SCAV-embed | 98 | 88 | 74 | 16 |
> | Mistral-7B-Instruct-RR | W/o Attack | 2 | 0 | 0 | 98 |
> |  | SCAV-embed | 94 | 84 | 70 | 20 |
>
> **[Rebuttal Table 7] Attacking LLMs with defense methods (ASR-keyword, %)**
>
> | Defense Methods | Advbench | StrongREJECT |
> | --- | --- | --- |
> | Self-Reminder [2] | 92 | 100 |
> | ICD [3] | 98 | 96 |
> | Paraphrasing [4] | 98 | 98 |
> | PPL [5] | 100 | 100 |
> | W/o Defense | 100 | 100 |
>
> [1] Improving Alignment and Robustness with Short Circuiting.
>
> [2] Defending chatgpt against jailbreak attack via self-reminders.
>
> [3] Jailbreak and guard aligned language models with only few in-context demonstrations.
>
> [4] Baseline defenses for adversarial attacks against aligned language models.
>
> [5] Detecting language model attacks with perplexity.

---

> > ### Author Response · Authors · 2024-08-13
> >
> > Dear Reviewer mJhn,
> >
> > Thank you for your thoughtful feedback on our submission. We kindly remind you to review our rebuttal and let us know if it adequately addresses your concerns. If you believe our explanations and revisions have satisfactorily resolved the issues, we would greatly appreciate it if you could reconsider your evaluation of our paper.
> >
> > Thank you again for your time and for providing valuable guidance. If there are any further questions or suggestions, we are glad to discuss them with you.
> >
> > Best regards,
> >
> > Authors of paper 7016

---

### Official Review · Reviewer_CRgs · 2024-07-13

**Soundness:** 3
**Presentation:** 4
**Contribution:** 3
**Rating:** 5
**Confidence:** 4

**Summary:**

This paper proposes a jailbreak attack (SCAV) inspired by the Concept Activation Vector (CAV) on neural networks. For safety concept, this concept vector is essentially defined as a direction orthogonal to the decision boundary of a linear classifier trained to distinguish safe and harmful instruction on embeddings at a given layer. SCAV works by perturbing the embedding in the direction of this concept vector (towards malicious and away from safe instructions).

Under the assumption that the concept “score” is linear, perturbing in concept vector direction is optimal. However, for token-level SCAV, the same technique does not apply so the authors come up with an optimization objective inspired by the embedding-level approach and use AutoDAN’s genetic algorithm to optimize it. Empirical results are convincing at both levels and show some good transferability to GPT-4.

**Strengths:**

### Originality

There are several works like JRE and RepE that attempt to mount a jailbreak attack at the embedding levels using an interpretability technique. However, this work fixes an obvious problem of these methods with a simple technique. It is also well-motivated by a classical interpretability method (albeit with a very strong assumption).

### Quality

All the formulations in Section 2 are technically sound given the linearity assumption. Algorithm 1 also makes a lot of sense. The experiments are relatively thorough (see the Weaknesses section for some suggestions) with a good number of baselines, different target models, transferability results, and some ablation studies. Among the other sections, I find Section 2.3 particularly clear and convincing.

### Significance

Strong and efficient jailbreak attacks help safety evaluation. This is a significant and timely research problem in my opinion. I also like the fact that this attack algorithm seems to also help us learn about the concept activations in LLMs (such as their linearity). The technique itself can also apply to other use cases beyond safety (concept erasure or unlearning is one other use case explored briefly in this paper).

### Clarity

The paper is well-written and easy to follow in all of the sections. I like the paper structure as well as how all of the ideas and results are presented.

**Weaknesses:**

### 1. Choice of layer to compute SCAV objective

I would like to see an ablation study on the choice of layer being optimized. Perhaps, Figure 4(a) does cover a bit of this, but I have a few more questions.

1. Why is only the last layer optimized in the token-level attack (Eq. (5)), unlike the embedding-level where the objective covers the prior layers? From Eq. (5), it is possible to optimize over $e_S^l$ while the objective is computed on any other layer $e_S^{k}$ where $k > l$. There seem to be some design choices here that were not explored. Presumably, if the objective depends on the last layer anyway, including earlier layers in the objective should not increase the overall computation.
2. What would happen if only the last layer is perturbed in the embedding-level attack? Is there a benefit from optimizing many layer from an early one to the last?
3. For both attacks, I would like to see how ASR changes with the choice of layer, given that only one layer is perturbed.

### 2. Token-level SCAV objective

I’m curious why the authors decide to use the objective in Eq. (5) instead of the Lagrangian relaxation of the optimization problem in Eq. (3) into something like the following:

$$
\arg\min_{e^L_S}~ \lvert| e^L_S - e^L \rvert| + \lambda P_m(e^L_S)
$$

Is there an advantage to the product form instead of a weighted sum?

### 3. Missing baseline attacks

1. **Comparison to soft prompt attack for embedding-level SCAV.** While the authors already compare SCAV to JRE and RepE, I believe it is also important to compare to other soft prompt optimization attack (such as https://arxiv.org/abs/2402.09063) as the concept is very similar. These soft prompts (can be prefix or suffix) are also known to be very successful against open-source LLMs.
2. **Comparison to GCG in Table 3 & 4 for token-level SCAV.** Unless there is a very good reason, I believe that it is important to have GCG as a baseline for any token-level attack, even if AutoDAN claims to be better than GCG in some scenarios. GCG is still the most popular attack at this point so it allows for easy comparison / verification across different works.

**Questions:**

**Q1:** For the embedding-level SCAV, when an embedding $e^l$ is mentioned (e.g., L115), does this refer to (1) embedding of one of the prompt tokens or (2) a newly added prefix (like soft prompt tuning) or suffix (like GCG but soft) token? I just want to make sure I understand the detail here, but it doesn’t seem clearly defined in Section 2.1.

**Q2:** This might be a follow-up question to Q1. I’m not very convinced by the statement on L121-122:

> The first term that minimizes $|\epsilon|$ ensures a small performance loss of LLMs, avoiding flaws such as repetitive or irrelevant responses.
>

unless the optimized embedding $e^l$ is from one of the prompt tokens. So I assume that $e^l$ is the next-token embedding (i.e., from the last input token)?

**Q3:** Can the authors share details on how Figure 2 is created? Or is it more like a diagram?

**Q4:** What’s the difference between ASR-answer and ASR-useful?

**Q5:** In Table 7 (Section 4.2), Is the SCAV attack here at embedding or prompt level? Since the baselines are GCG and AutoDAN, I’m assuming that this is at a prompt level?

**Limitations:**

Limitations and negative societal impact have been adequately addressed.

---

> ### Author Rebuttal · Authors · 2024-08-05
>
> Thanks for your insightful comments and questions.
>
> > **Weakness 1: Optimizing only the last layer in token-level attack**
> >
>
> We optimize only the last layer to ensure
>
> 1. **A fair comparison with baselines**: our baselines (e.g., [1]) use only information from the last layer so we follow their setting to ensure a fair comparison.
> 2. **Simplicity and effectiveness**: using only the last layer is 1) **simple** - does not require balancing different layers of varying embedding scales and 2) **effective** - better than baselines and achieves a **comparable performance** with a method considering multiple layers, as shown in Rebuttal Table 1.
>
> **[Rebuttal Table 1] Comparison with SCAV-m, which considers multiple layers by extending Eq. (5) to** $\arg\min_{S} \max_{l}[P_m(e_S^l) ||e_S^l - e^l||],\text{TestAcc}(P_m^l)>P_1$ **on Advbench**
>
> | Model | Method | ASR-keyword (%) | ASR-answer (%) | ASR-useful (%) | Language Flaws (%) |
> | --- | --- | --- | --- | --- | --- |
> | LLaMA-2-7B-Chat | SCAV | 54 | 60 | **44** | **52** |
> |  | SCAV-m | **60** | 60 | 34 | 58 |
> | LLaMA-2-13B-Chat | SCAV | **72** | 46 | 28 | 58 |
> |  | SCAV-m | 64 | **50** | **38** | **48** |
>
> > **Weakness 2: What would happen when perturbing only the last layer**
> >
>
> Perturbing only the last layer **results in worse ASRs and more language flaws**, as shown in Rebuttal Table 2. We suspect it is because, at the last layer, the inference procedure has nearly finished, thus forming a completely different answer can be difficult.
>
> **[Rebuttal Table 2] Comparison with SCAV-l that perturbs only the last layer**
>
> | Dataset | Methods | ASR-keyword (%) | ASR-answer (%) | ASR-useful (%) | Language Flaws (%) |
> | --- | --- | --- | --- | --- | --- |
> | Advbench | SCAV | **100** | **96** | **92** | **2** |
> |  | SCAV-l | 18 | 4 | 2 | 82 |
> | StrongREJECT | SCAV | **100** | **98** | **96** | **10** |
> |  | SCAV-l | 28 | 14 | 2 | 64 |
>
> > **Weakness 3: Impact of each layer on ASR**
> >
>
> We provide the results in the rebuttal PDF attached to the global rebuttal. Our findings are:
>
> 1. For both levels, **ASR is poor on layers that are not linearly separable** (Figure 2 in the rebuttal PDF).
> 2. For **embedding-level** attacks, perturbing the last layer is less effective compared with some earlier layers (e.g., layers 13 to 23). Algorithm 1 tends to select these effective (Figure 2 in the PDF).
> 3. For **prompt-level** attacks, using middle or late layers has a comparable attack performance (Figure 3 in the PDF).
>
> > **Weakness 4: Why product form**
> >
>
> We use the product form because it works sufficiently well (Table 3) **without introducing an additional hyperparameter $\lambda$** for balancing $||e^L_S-e^L||$ and $P_m$, which is required for the weighted sum expression. In the product form, the percentage of increase in $||e^L_S-e^L||$ is considered to be similarly important to the percentage of increase in $P_m$, without having to consider their difference in scales.
>
> > **Weakness 5: Comparison to soft prompt attack [2]**
> >
>
> As shown in Rebuttal Table 3, our embedding-level attack method outperforms the soft prompt attacks in terms of ASRs and language flaws.
>
> **[Rebuttal Table 3] Comparing with soft prompt attacks**
>
> | Evaluation Dataset | Models | Methods | ASR-keyword (%) | ASR-answer (%) | ASR-useful (%) | Language Flaws (%) |
> | --- | --- | --- | --- | --- | --- | --- |
> | Advbench | LLaMA-2 (7B-Chat) | SCAV | 100 | 96 | 92 | 2 |
> |  |  | soft prompt | 56 | 50 | 40 | 62 |
> |  | LLaMA-2 (13B-Chat) | SCAV | 100 | 98 | 96 | 0 |
> |  |  | soft prompt | 80 | 66 | 50 | 44 |
> | StrongREJECT | LLaMA-2 (7B-Chat) | SCAV | 100 | 98 | 96 | 10 |
> |  |  | soft prompt | 64 | 44 | 38 | 66 |
> |  | LLaMA-2 (13B-Chat) | SCAV | 100 | 100 | 98 | 2 |
> |  |  | soft prompt | 74 | 28 | 28 | 68 |
>
> > **Weakness 6: Comparison to GCG**
> >
>
> Results show that our prompt-level attack method consistently outperforms GCG on different datasets and tasks (direct prompt-level attack in Rebuttal Table 4 and transferability in Rebuttal Table 5 of the rebuttal PDF).
>
> **[Rebuttal Table 4] Comparing with GCG**
>
> | Evaluation Dataset | Models | Methods | ASR-keyword (%) | ASR-answer (%) | ASR-useful (%) | Language Flaws (%) |
> | --- | --- | --- | --- | --- | --- | --- |
> | Advbench | LLaMA-2 (7B-Chat) | SCAV-prompt | 54 | 60 | 44 | 52 |
> |  |  | GCG | 28 | 32 | 10 | 76 |
> |  | LLaMA-2 (13B-Chat) | SCAV-prompt | 72 | 46 | 28 | 58 |
> |  |  | GCG | 40 | 24 | 10 | 74 |
> | StrongREJECT | LLaMA-2 (7B-Chat) | SCAV-prompt | 60 | 46 | 40 | 44 |
> |  |  | GCG | 26 | 26 | 16 | 72 |
> |  | LLaMA-2 (13B-Chat) | SCAV-prompt | 54 | 48 | 46 | 42 |
> |  |  | GCG | 34 | 18 | 16 | 80 |
>
> > **Questions 1 and 2: What does $e^l$ refers to?**
> >
>
> $e^l$ is the embedding of the last input token. Specifically, given an input instruction with N tokens, we first attack the embedding of the N-th token to generate the first token in the answer. We then iterate this process to generate a full answer.
>
> > **Question 3: How is Figure 2 created**
> >
>
> Figure 2 is generated by 1) generating two-dimensional pseudo-embedding data points and 2) running the attacking methods (ours, RepE, and JRE) to obtain the perturbation vector.
>
> > **Question 4: Difference between ASR-answer and ASR-useful**
> >
>
> ASR-useful is a more strict criterion than ASR-answer. **ASR-answer** mainly evaluates whether the model answers or rejects to answer the question, while **ASR-useful** evaluates whether the model answer is truly useful for attackers. For details, please refer to Appendix A.3.2 and F.
>
> > **Question 5: Is SCAV in Table 7 embedding or prompt level?**
> >
>
> It is embedding-level SCAV. The prompt-level baselines are added following [3].
>
> [1] AutoDAN: Interpretable Gradient-Based Adversarial Attacks on Large Language Models
>
> [2] Soft Prompt Threats: Attacking Safety Alignment and Unlearning in Open-Source LLMs through the Embedding Space
>
> [3] Eraser: Jailbreaking Defense in Large Language Models via Unlearning Harmful Knowledge

---

> > ### Author Response · Authors · 2024-08-13
> >
> > Dear Reviewer CRgs,
> >
> > Thank you for your thoughtful feedback on our submission. We kindly remind you to review our rebuttal and let us know if it adequately addresses your concerns. If you believe our explanations and revisions have satisfactorily resolved the issues, we would greatly appreciate it if you could reconsider your evaluation of our paper.
> >
> > Thank you again for your time and for providing valuable guidance. If there are any further questions or suggestions, we are glad to discuss them with you.
> >
> > Best regards,
> >
> > Authors of paper 7016

---

### Official Review · Reviewer_6xey · 2024-07-15

**Soundness:** 3
**Presentation:** 3
**Contribution:** 3
**Rating:** 6
**Confidence:** 4

**Summary:**

This paper introduces a new framework named Safety Concept Activation Vector (SCAV) for jailbreaking LLMs. This framework is built upon LLM interpretability work. Specifically, the paper utilizes an interoperability approach called Concept Activation Vector, which can linearly separate safe and unsafe instructions in the latent representation space. Given this linear separability, this paper proposes that a jailbreak attack can be achieved if one can make perturbations such that the latent feature of an unsafe prompt is perturbed to the feature subspace of a safe prompt. Following this idea, the authors show the feasibility of doing such jailbreak attacks in both the feature space and the prompt space.

**Strengths:**

1. The paper is well-written and well-motivated. The presentation is also clean and comprehensive.
2. The success rate of the attack is good, showing good improvement over existing baselines. The improved transferability to attack GPT-4 is also an advantage of the attack.
3. The paper also shows that existing unlearn methods fail to defend against the proposed attack, suggesting that unlearning may not be an effective solution to stronger jailbreaking attacks.

**Weaknesses:**

1. The idea of using an interpretability approach to assist jailbreak attacks is not new and has already been explored by [1].
2. It would be better if the authors could also discuss why the proposed attack can be more effective than previous ones and what are possible ways to mitigate the attacks.

[1] Zou, Andy, et al. "Representation engineering: A top-down approach to ai transparency." arXiv preprint arXiv:2310.01405 (2023).

**Questions:**

Can the authors clarify more on how the prompt space attack is implemented? In the paper, the authors only mentioned the use of AutoDAN’s hierarchical genetic algorithm to solve the optimization. It would be better if more details could be clarified.

For the representation space attack, the adversaries can directly interfere with the internal components of the model. Would this be closer to the threat model of fine-tuning attacks presented in [1]? In that case, it seems that fine-tuning attacks are easier and more effective. It would be good if the authors could discuss the connection.

[1] Qi, Xiangyu, et al. "Fine-tuning aligned language models compromises safety, even when users do not intend to!." arXiv preprint arXiv:2310.03693 (2023).

**Limitations:**

No major limitations were found.

---

> ### Author Rebuttal · Authors · 2024-08-04
>
> Thanks a lot for your insightful and constructive comments. We really appreciate the efforts that you have made in helping improve our paper.
>
> > **Weakness 1: Interpretability used in attacks by [1].**
> >
>
> We acknowledge that previous works like [1] have explored the use of interpretability to assist attacks. In contrast, our work distinctively focuses on developing a **more accurate and principled interpretation method**. Unlike existing approaches, our method:
>
> 1. Eliminates misleading heuristics (L128-140) and thus performs significantly better than existing techniques [1,2] (Tables 1 and 2), particularly under conditions of limited training data (Figure 3).
> 2. Is the first to support both embedding-level and prompt-level attacks, enhancing its applicability.
> 3. Removes the necessity for time-consuming hyperparameter tuning.
>
> We will update our abstract and introduction to better highlight these distinctions.
>
> > **Weakness 2: Need to discuss why the proposed attack is more effective**
> >
>
> Why our method is more effective is introduced in the method part (L128-140, L158-165) and we will discuss more in the experiments.
>
> 1. For embedding-level attacks, our method is more effective because we **remove potentially misleading heuristics** of existing methods (L128-140). While existing works **assume** that some heuristically extracted direction (e.g., the main component of PCA) works well for perturbing the embedding without theoretical justification, we **learn** the malicious probability of each embedding from data to provide an accurate perturbation.
> 2. For prompt-level attacks, our method **can more accurately estimate the attack success rate** because, unlike the existing works, we do not rely on manually defined target responses, which are often different from the real model responses (more details in L158-165).
>
> > **Weakness 3: How to mitigate the attacks**
> >
>
> Mitigating attacks is important and we will add the following discussions in the revised version:
>
> 1. **Applying existing defense methods cannot effectively mitigate our attacks**. As shown in Rebuttal Table 1, after applying existing defense methods [3,4,5,6], our attack method still achieves a high attack success rate (ASR) on two datasets according to the widely used criterion ASR-keyword (%). Here, the victim LLM is LLaMA-2 (7b-Chat). This indicates that our work identifies the inherent safety vulnerability of LLMs. Moreover, the unlearn method in Section 4.2 can also be regarded as a defense method and it also fails to mitigate our attacks (Table 7). We also attack models with adversarial training proposed by [8](Rebuttal Table 2).
>
>     **[Rebuttal Table 1] The attack success rate** **(ASR-Keyword in %) after applying defense methods**
>
>     | Defense Method | On Advbench | On StrongREJECT |
>     | --- | --- | --- |
>     | ICD [3] | 98 | 96 |
>     | Self-Reminder [4] | 92 | 100 |
>     | Paraphrasing [5] | 98 | 98 |
>     | PPL [6] | 100 | 100 |
>     | W/o Defense | 100 | 100 |
>
>     **[Rebuttal Table 2] Attacking LLMs with adversarial training [8] on Advbench**
>     | Models  | ASR-keyword (%) | ASR-answer (%) | ASR-useful (%) | Language Flaws (%) |
>     | --- | --- | --- | --- | --- |
>     | LLaMA-3-8B-Instruct-RR | 98 | 88 | 74 | 16 |
>     | Mistral-7B-Instruct-RR | 94 | 84 | 70 | 20 |
>
> 2. **Suggesting other ways for mitigating the attacks**. One potential way to mitigate the attacks is to distinguish perturbed embeddings from original ones, based on the assumption that some perturbed embeddings might be different from original embeddings. We are eager to explore whether such an assumption is valid in the future.
>
> > **Question 1: Implementation detail of prompt space attack**
> >
>
> Per your suggestion, we will add the following details to ensure that our paper is self-contained:
>
> “Specifically, the hierarchical genetic algorithm is tailored for structured prompt text. It views the jailbreak prompt as a combination of paragraph-level population and sentence-level population. At each search iteration, it first optimizes the sentence-level population by evaluating and updating word choices within sentences. Then, it integrates these optimized sentences into the paragraph-level population and performs genetic operations to refine sentence combinations, ensuring comprehensive search and improvement across both levels with high jailbreak performance and readability.”
>
> > **Question 2: Comparing with fine-tuning attacks [7]**
> >
>
> Per your suggestion, we compare with [7] and will add related results in the paper. As shown in the following tables, our SCAV embedding-level attack:
> 1. achieves a **higher attack success rate** (ASR-keyword) compared with [7] (Rebuttal Table 3)
> 2. requires **much less time cost and computation memory** when applying on LLaMA-2-7B-Chat, since it only requires inference (Rebuttal Table 4). The evaluation is conducted on Advbench.
>
> **[Rebuttal Table 3] Comparison with fine-tuning attacks in terms of ASR-Keyword (%)**
>
> | Victim LLM | Ours | fine-tuning attack |
> | --- | --- | --- |
> | LLaMA-2-7B-Chat | 99.2 | 95.6 |
> | GPT-3.5-turbo | 95.7 | 85.0 |
>
> **[Rebuttal Table 4] Comparison with fine-tuning attacks in terms of efficiency**
>
> |  | Ours | fine-tuning attacks |
> | --- | --- | --- |
> | Time cost | 30 s | 2 min |
> | Computation memory | 15 G of one A100 GPU | 80 G of two A100 GPUs |
>
> [1] Representation engineering: A top-down approach to ai transparency
>
> [2] Open the Pandora's Box of LLMs: Jailbreaking LLMs through Representation Engineering
>
> [3] Jailbreak and guard aligned language models with only few in-context demonstrations
>
> [4] Defending chatgpt against jailbreak attack via self-reminders
>
> [5] Baseline defenses for adversarial attacks against aligned language models
>
> [6] Detecting language model attacks with perplexity
>
> [7] Fine-tuning aligned language models compromises safety, even when users do not intend to!
>
> [8] Improving Alignment and Robustness with Short Circuiting

---

> > ### Comment · Reviewer_6xey · 2024-08-09
> > **Thanks Authors**
> >
> > I would like to thank the authors for their detailed responses. I believe this paper makes a good improvement of LLM jailbreak attacks, which could be used to further encourage future development of stronger defenses. Thus, I will keep my initial recommendation of weak accept.
> >
> > Meanwhile, I noticed the authors repeatedly use the term "misleading heuristics" throughout the paper and the rebuttal context. This term is very hand-wavy. I suggest the authors be more precise on what this means, which may make the audience easier to understand.
> >
> > I also noticed that the authors reported higher ASR than fine-tuning attacks. Can the authors share their intuition on why an embedding-level attack would be even stronger than fine-tuning that directly modifies model weights?

---

> > > ### Author Response · Authors · 2024-08-10
> > > **Thanks for your comment!**
> > >
> > > Thanks for your comment!
> > >
> > > **Clarifying the specific meaning of the term “misleading heuristic”**
> > >
> > > Thanks for pointing out the potential confusion caused by using this term. We are glad to provide a clearer and more factual explanation of what this term means.
> > >
> > > - We observed the linear separability in embeddings (Figure 1) and the relationship between jailbreak ASR and $P_m$ (Table 9), which guided the perturbation vector learning. In comparison, the extraction methods used by RepE and JRE are assumed heuristically.
> > > - We found that RepE and JRE sometimes cannot correctly extract the vector direction as described by their heuristic (Figure 2). Moreover, in a small number of training sample scenarios, these two baselines result in a lower ASR (Figure 3) because of this.
> > >
> > > Following your suggestion, we will provide more precise explanations in our paper, instead of just using this term to summarize.
> > >
> > > **Intuition on why SCAV-embedding outperforms fine-tuning attack**
> > >
> > > We are glad to share our intuition on why SCAV-embedding performs better than fine-tuning attacks with you.
> > >
> > > - Fine-tuning directly modifies the weight of models, which cannot adaptively modify their behavior towards specific malicious instructions. However, our paper's Algorithm 1 can ensure accurate and effective attacks by adaptively modifying the embeddings of each layer.
> > > - LLMs obtain safety concepts through a large number of corpus fine-tuning after pre-training. A small amount of corpus used for fine-tuning attacks may not be generalized enough to bypass all the capabilities of the safety concepts. However, SCAV-embedding attacks directly modify the safety concepts in embeddings to reverse LLMs’ recognition of malicious instructions to safe ones, thus leading to higher ASR.
> > >
> > > Thank you for praising our work in the comments. We hope the above content can clarify your doubts. We are glad to continue to provide you with any details.

---

### Author Rebuttal · Authors · 2024-08-06

Thanks to all the reviewers for the insightful comments and recognition of our work. We are encouraged that reviews think our paper has the following strengths:

- Well-written and easy to follow (Reviewers 6xey, CRgs, xmcv)
- Comprehensive experiments (Reviewers 6xey, CRgs, xmcv)
- Superior performance (Reviewers 6xey, CRgs, mJhn, xmcv)
- Insightful (Reviewers 6xey, CRgs, xmcv)
- Fix some problems of existing methods (Reviewers 6xey, CRgs)

Following are some of the main concerns, afterwards we address each reviewer's comments individually.

- **To Reviewer 6xey**: We greatly appreciate your suggestions, which have indeed made our paper more readable and accurate. We will clearly and precisely describe the advantages of the SCAV attack compared to baselines such as RepE and JRE. We will also provide a more comprehensive evaluation and experimental results to solidify our findings and highlight our significant advancements.
- **To Reviewer CRgs**: Thank you for your suggestions on several critical parts of our paper. We are pleased to discuss some of the key design decisions we made and will summarize these design ideas in the appendix to provide readers with more detailed information. Per your suggestions, we conducted additional comparative experiments, which confirmed the validity of our design choices. These experimental results have strengthened the solidity of our paper.
- **To Reviewer mJhn**: We appreciate your insightful comments. We fully understand your concerns about the key assumption of the linear separability of safety concepts. Following your advice, we repeated the experiments on a broader set of models and datasets, which still support our findings as stated in the paper. Additionally, we have elaborated on the benefits of using probabilities for perturbation and provided detailed explanations for choosing $P_0$ and $P_1$, which may address your concerns.
- **To Reviewer xmcv**: Thank you for your important suggestions, which have allowed us to enhance the focus on particularly interesting aspects of our work. We repeated the experiments you were skeptical about, expanded the evaluation metrics to include weighted bi- and tri-gram entropy, and provided more findings and explanations for the related results. We also appreciate your concern regarding the ethical compliance of the human involvement. We have obtained IRB-equivalent approval, which will be explicitly stated in the paper.

To provide you with some key experimental results more intuitively, we have included an additional PDF with this rebuttal. Once again, thank you for your time and your invaluable contribution to the community!

---

### Decision · Program_Chairs · 2024-09-25

**Decision:**

Accept (poster)

**Comment:**

This paper proposes a new jailbreak attack based on a Safety Concept Activation Vector (SCAV) framework, which effectively guides the attacks by accurately interpreting LLMs' safety mechanisms. The authors have put a lot of effort into addressing the reviewers' concerns by conducting massive new experiments and detailed clarifications.

Although the last two reviewers maintain their initial negative scores, it seems that the detailed rebuttal has well addressed their concerns. Specifically, the Reviewer mjhn has not relied at all during the discussion phase. But it is clear to judge that the authors have well addressed their comments (about more experiments and ablations). The Reviewer xmcv persisted in theoretical analyses of the proposed method, which the AC thinks is not necessary for a paper to be accepted at NeurIPS. The authors have thoroughly validated the superiority of their method with extensive explanations and experiments ("Beyond the 2 datasets and 8 LLMs already tested, we have extended our experiments to Harmbench and 3 more LLMs with different training paradigms"), which the other three reviewers have appreciated.

The authors should incorporate the new results into their final version as promised. In addition, since most reviewers are concerned about the novelty, it makes sense to explicitly describe the differences between the proposed attack and other interpretability-based attacks.
Finally, defenses that are tailored to the proposed attacks should be included, e.g., following the statement "One potential way to mitigate the attacks is to distinguish perturbed embeddings from original ones, based on the assumption that some perturbed embeddings might be different from original embeddings."